# Identification of Emerging Roadkill Hotspots on Korean Expressways Using Space–Time Cubes

**DOI:** 10.3390/ijerph20064896

**Published:** 2023-03-10

**Authors:** Minkyung Kim, Sangdon Lee

**Affiliations:** Department of Environmental Science and Engineering, Ewha Womans University, Seoul 03760, Republic of Korea; enviecol@ewha.ac.kr

**Keywords:** spatiotemporal, wildlife–vehicle collision, priority, reduction measure, water deer

## Abstract

Collisions with wild animals on high-speed expressways not only lead to roadkill but can also cause accidents that incur considerable human and economic costs. Based on roadkill data from 2004–2019 for four common wildlife species involved in collisions with vehicles on expressways in Korea (water deer, common raccoon dog, Korean hare, and wild boar), the present study conducted optimized hotspot analysis and identified spatiotemporal patterns using a space–time cube (STC) approach. Temporal and spatial differences in the roadkill data were observed between species. Water deer were the most common roadkill species of the four studied, with hotspots in the southern region of the capital area, in the Chungnam region, and in the western Chungbuk and Gangwon-do regions. However, the instances of water deer roadkill over time differed between each region. In addition, it was found that the number of cases of wild boar roadkill has increased recently. In particular, a number of new hotspot areas were observed centered on the metropolitan area Gyeonggi-do, which contains a high population and significant infrastructure. Overall, the emerging hotspot analysis based on STCs was able to determine cold spot and hotspot trends over time, allowing for a more intuitive understanding of spatiotemporal clustering patterns and associated changes than cumulative density-oriented hotspot analysis. As a result, it becomes easier to analyze the causes of roadkill and to establish reduction measures according to priority.

## 1. Introduction

The length of the road network in Korea has continuously increased in recent decades [1]; as of 2019, the total length of roads in Korea totaled 111,314 km, consisting of 4767 km of expressways (4.3%), 14,030 km of general national highways (12.6%), 4945 km of special city and metropolitan city roads (4.4%), 18,047 km of local roads (16.2%), 30,307 km of city roads (27.2%), 22,776 km of county roads (20.5%), and 16,442 km of district roads (14.8%), [2]. Expressways were first introduced to Korea with the completion of the Gyeongin Expressway in 1968 [3], providing high-speed transportation for vehicles and acting as the backbone of the transportation network connecting major cities. As of 2019, expressways accounted for 71.8% of the daily traffic volume [4]. In addition, from 2000, the total length of roads increased by 1.3% per year, while the expressways increased rapidly by 6.5% annually.

Road expansion is the major cause of land fragmentation [5], in which natural habitats are divided into smaller areas, where the total habitat area decreases. This can lead to a gradual decrease in wildlife populations due to inbreeding and a lack of natural resources [6], leading to a loss of species and deterioration in ecosystem functions [7]. In addition, wildlife–vehicle collisions (WVCs), several of which lead to roadkill, commonly occur when a wild animal attempts to cross a road within its natural range. Medium-sized and large mammals such as water deer (*Hydropotes inermis*) and wild boar (*Sus scrofa*), which require a relatively larger area compared to small mammals, are in danger of WVCs due to a lack of habitat [8].

WVCs not only pose a direct threat to wildlife species but can also lead to human injury and death and material damage; thus, their management is very important. Accordingly, the Korea Expressway Corporation (KEC), which manages Korea’s expressways, has recorded all cases of roadkill on expressways from 2004 to the present, including the species involved, spatial data such as the exact point and route, and temporal data such as the date and time. In addition, to establish comprehensive measures to prevent WVCs, the KEC conducted a detailed ecological survey in 2008 and research on facility evaluation and the development of alternatives [9,10]. As a result, the annual number of roadkill cases found on expressways in Korea has decreased, but, nevertheless, thousands of WVCs still occur annually, and there are differences in the roadkill incidence rates and hotspot areas between species [1].

In Korea, the first study on WVCs analyzed the characteristics of roadkill cases on a central highway [11], and various influencing factors such as species, year, season, and land use have subsequently been analyzed [12,13]. In addition, a roadkill hotspot study was conducted using spatial analysis [14], while roadkill clusters have also been investigated [15]. Recently, Kim et al. [1] analyzed the temporal and spatial trends of roadkill on highways across Korea over the past 15 years to determine priority roadkill mitigation measures. However, all of these studies targeted temporal or spatial data separately, and thus, could not accurately reflect spatiotemporal changes in roadkill patterns. Because WVCs have both spatial and temporal characteristics in terms of the location and the date of the accident, respectively, spatiotemporal analysis must be conducted [16].

The identification of WVC hotspots is crucial for traffic safety management, and an understanding of the trends in WVCs can help inform policy decisions designed to prevent these collisions and mitigate associated damage [17]. In this respect, the space–time cube (STC) model proposed by Hagerstrand [18] is a useful method of analysis that can consider the temporal dimension on a spatially two-dimensional plane [19] and has been -widely used in traffic safety, public health, and other fields [20,21,22]. Because spatiotemporal data can be aggregated while maintaining continuity using this method [23,24], it is useful for the spatiotemporal analysis of WVCs, making it possible to intuitively identify areas of low risk and provide information that can be used as the basis for establishing prevention and control strategies [24]. Nevertheless, few studies using hotspot analysis based on the STCs have been carried out to analyze the WVCs occurrence characteristics.

The present study identifies spatiotemporal roadkill patterns through STC-based analysis for four main wildlife species from 2004 to 2019 on expressways in Korea. In particular, it determines cold spots and hotspots for each species, with the aim to provide information that can be used to devise measures to reduce WVCs.

## 2. Materials and Methods

This study analyzed roadkill patterns on expressways managed by the KEC in Korea (Figure 1). The KEC employees in charge of roadkills record data, detailed information including dates, times, longitude and latitude, the number and type of carcasses discovered during daily expressway patrols in three shifts on a daily basis regularly. Thereafter, they remove the animal carcasses from the road to ensure that the same carcass is not recorded twice [25].

Overall, 36,863 cases of roadkill were recorded by the KEC from 2004 to 2019. The species most frequently involved was water deer with 28,045 cases, followed by the common raccoon dog (*Nyctereutes procyonoides*) with 5246 cases, the Korean hare (*Lepus coreanus*) with 1511 cases, and wild boar with 749 cases, though wild boar were only recorded from 2011 [1]. Overall, approximately 96.44% of the 36,863 animals killed on expressways in Korea were from these four species; thus, they were targeted for subsequent hotspot analysis. We analyzed both the analysis of the optimized hotspot and the emerging hotspot of the space-time cube in ArcGIS Pro.

First, optimized hotspot analysis was conducted to determine the areas with a high roadkill density. In this way, it was possible to analyze hotspots and cold spots based on the number of roadkill occurrences and to identify regions with statistically significant patterns. For optimized hotspot analysis, the individual roadkill cases within a specific management boundary or grid cell were combined and analyzed. For this hotspot analysis, the grid cell size was set at 4 × 4 km, and the Getis–Ord Gi* statistic was employed [26] for 90%, 95%, and 99% confidence intervals (Equation (1)).
(1)$$G_{i}=\frac{\sum_{j=1}^{n}w_{i,j}x_{j}-\bar{X}\sum_{j=1}^{n}w_{i,j}}{S\sqrt{\frac{n\sum_{j=1}^{n}w_{i,j}^{2}-{\left(\sum_{j=1}^{n}w_{i,j}\right)}^{2}}{n-1}}}$$
where *x_j_* is the attribute value for feature *j*, *w_i,j_* is the weighted value of the space between feature *i* and *j*, and *n* is the total number of features.

The Gi* statistic is expressed as a z-score indicating statistically significant hotspots and cold spots, with values of >2.58, 1.96–2.58, and 1.65–1.96 representing hotspots or cold spots with confidence levels of 99%, 95%, and 90%, respectively.

Furthermore, emerging hotspot analysis was conducted to examine the temporal changes in roadkill occurrences over the 16-year study period. For this, it was first necessary to create a three-dimensional STC at a specific location based on the temporal trend in the roadkill data (Figure 2). The time series of the z-scores for that location was analyzed using the Mann–Kendall statistic. This analysis returns the clustering trend z-score, *p*-value, and binning category for each location. The time axis for the STCs was set to 1 year in order to examine the annual roadkill trends over the 16-year period. However, because the STC approach requires at least 10 individual time periods, the time axis for wild boar employed 6-month time intervals. The identified hotspots and cold spots were classified into eight types each based on their following temporal patterns: new, consecutive, intensifying, persistent, diminishing, sporadic, oscillating, and historical (Table 1).

## 3. Results

### 3.1. Optimized Hotspot Analysis by Animal Species

Hotspot analysis was conducted on 36,863 cases of roadkill that occurred over 16 years on Korean expressways, identifying those regions with high or low roadkill numbers. In Figure 3, Figure 4, Figure 5 and Figure 6, the darker the red (hotspot) or blue (cold spot) region, the less likely that the clustering was the result of chance. White represents statistically insignificant areas (i.e., no clustering patterns).

Figure 3a presents the optimized hotspot results for water deer roadkill, which accounted for 76.08% of the 36,863 roadkill cases. Significant hotspots were observed in the southern part of the capital area, around the Chungnam region, in the western part of Chungbuk, and in the western region of Gangwon province. In contrast, significant cold spots were found in the western part of the capital area, Jeonnam province, South Gyeongbuk province, and the Gyeongnam area. Raccoon dog showed the optimized hotspot in Gangwon, Jeonnam, and Jeonbuk areas (Figure 4a), and the Korean hare showed the optimized hotspot only in the Gangwon area (Figure 5a). In addition, since 2011, 778 wild boar roadkill have accounted for 2.03% of the total roadkill numbers, with hotspots in the eastern region of Chungbuk, and parts of Chungnam, Jeonbuk, and Gyeongbuk (Figure 6a).

### 3.2. Spatiotemporal Analysis of Roadkill by Species

As a result of the spatiotemporal analysis of roadkill accidents by species on expressways from 2004 to 2019, a total of 185 hotspot regions were identified for water deer, consisting of 49 consecutive, 41 persistent, 1 Diminishing, 9 sporadic, 78 oscillating, and 7 historical hotspots. There was a total of 500 cold spots for water deer, consisting of 4 new, 36 consecutive, 24 intensifying, 20 persistent, 20 diminishing, 339 sporadic, and 57 oscillating cold spots (Table 2 and Figure 3b).

Looking at Gangwon province and the capital area, cumulative road-kill hotspots for 16 years are widely distributed, as shown on the left of Figure 7. However, the result of classifying into subdivided types according to the time trend is shown on the right of Figure 7, and the types can be divided into persistent or intensifying or consecutive hotspot sections and so on. Persistent hotspots (those that are present without an increase or decrease over at least 90% of the individual time periods) were most common on the Yeongdong Expressway, which connects the southeast of the capital area to the western area of Gangwon. This expressway was also associated with consecutive hotspots, which are defined as areas with continuously high numbers of WVCs in recent years. Therefore, it is necessary to prioritize reduction measures for such persistent hotspots or intensifying hotspots, which are hotspot areas that are gradually strengthened according to the trend of time, rather than other areas. In addition, a part of the capital area and the entire Chungnam area contained oscillating hotspots. Compared with the optimized hotspots (Figure 3a), the spatiotemporal analysis revealed that the roadkill frequency varied over time, even for the hotspots in the central region of Korea. Based on these results, it will be possible to establish priorities and response strategies to reduce the incidence of WVCs.

For raccoon dogs, hotspot analysis revealed no significant hotspot areas (Figure 4b and Table 2), while there were 625 cells representing cold spots, made up of 10 new, 44 consecutive, 65 sporadic, and 506 oscillating cold spots. Some parts of the capital area and some regions of Gyeongbuk and Gyeongnam were associated with consecutive cold spots. However, most recent cold spots were of an oscillating cold spot type, i.e., were a hotspot in the past. For example, Jeonbuk and Jeonnam province were classified as hotspots in the optimized hotspot analysis but were classified as oscillating cold spots based on recent occurrences (Figure 4).

There were 1513 cases of Korean hare roadkill, but as shown in Figure 5b, neither hotspots nor cold spots were identified using spatiotemporal analysis (Table 2), meaning that there were no statistically significant clusters in the study range during the study period. However, with the optimized hotspot analysis, hotspots were identified in the vicinity of Gangwon province.

Two types of statistically significant hotspot were found for wild boar using the 6-month intervals (Figure 6), with 49 new hotspots and 25 sporadic hotspots identified (Table 2). Significant new hotspots appeared within the capital area, some areas in Chungnam, and some areas in Gyeongbuk, where sporadic hotspots were also found. Thus, this spatiotemporal analysis can be used to identify priority areas for reducing the occurrence of wild boar roadkill, even in regions that were not classified as hotspots using optimized hotspot analysis.

## 4. Discussion

In this study, based on 16 years of roadkill data on high-speed expressways in Korea, we analyzed the spatiotemporal patterns of roadkill accidents using STCs and optimized hotspot/cold spot analysis for four major species. There were differences in the temporal and spatial occurrence of roadkill between these species. Water deer, the species with the highest number of roadkill cases, had hotspots in the southern part of the capital area, around the Chungnam region, in the western part of Chungbuk, and in the western part of Gangwon province. However, the pattern of occurrence over time differed for each region. In addition, all four species had hotspots in areas near the Jungang Expressway and Yeongdong Expressway in Gangwon-do. For raccoon dogs, hotspots were found in the Jeonbuk region, northern Jeonnam, and the western part of Gangwon-do; however, spatiotemporal analysis revealed that most of these were gradually decreasing or oscillating between hotspots and cold spots, while the Korean hare did not have any statistically significant hotspots in the spatiotemporal analysis. In contrast, there was an increase in the number of wild boar roadkill cases, with new hotspots found in several areas (the capital area and some areas in Chungnam and Gyeongbuk Province). In particular, a number of new hotspot areas were centered in the capital area (Gyeonggi-do), where the population and infrastructure are concentrated. The present study found that there were differences in the location of roadkill hotspots between animal species on expressways in Korea over the 16-year study period, which is likely to be related to the density of their populations and their habitat range [1]. Our results, which classify types in detail, help to identify the causes and identify priority areas for mitigation measures. In addition, hotspots were found in the vicinity of Jungang Expressway and Yeongdong Expressway, which are in the western part of Gangwon-do. Therefore, our results suggest that this area should be managed as a roadkill hotspot area regardless of species.

Given the severity of accidents involving WVCs, decision-makers and safety managers need to better understand the characteristics of WVCs and identify high-risk locations to mitigate the effects of these crashes, as well as appropriate responses to action should be prioritized. However, understanding the occurrence of roadkill is very difficult because it usually changes in both spatial and temporal dimensions. Moreover, many studies separately analyzed and discussed the spatial distribution and the temporal trend so far [1,27]. Previous studies have analyzed roadkill distributions to determine spatial hotspots, but these studies have not considered changes over time [28]. Therefore, in this study, we employed STCs that aggregate spatiotemporal information for the same data set to simultaneously and comprehensively analyze spatiotemporal information. This represents a new method for spatiotemporal analysis, meets the requirements of retention time and spatial continuity, and has the advantage of visualizing the analysis results [24,29]. In recent years, the emerging hot spot analysis based on STCs has gradually been applied to different scientific fields [30,31,32]. The results of the emerging hotspot analysis based on STCs offer important information on the locations and times of WVCs, allowing decision-makers to observe spatiotemporal clustering patterns. Thus, it is easier to grasp the causes of WVCs and establish measures according to priority to reduce them.

Recently, studies on the reliability and accuracy of roadkill data collected to implement appropriate roadkill mitigation measures are being conducted [33,34,35]. The potential biases of roadkill data could be due to (1) carcass removal, for example, by scavenging animals or people, (2) detectability, as small road-killed animals are less likely to be observed than are large animals and (3) injury; that is, animals are not immediately killed by the collision but are injured and die some distance, out of sight of the road [36,37]. This could lead to an underreporting of the actual number of deaths. In Korea, efforts are being made to reduce such under-reporting and redundant reporting through a system that collects data on a regular basis (3 shifts, 24 h). In addition, in this paper, in order to supplement this point, the analysis was conducted focusing on large animals. However, since the reduction of the size of the data is a very important issue in roadkill analysis, it is judged that additional research to compensate for this needs to be supplemented.

## 5. Conclusions

Korea’s highway management institutions, the KEC, and the Ministry of Environment, aim to reduce roadkill using reduction measures such as ecological passages, warning signs, and guided fences. It would be best to establish roadkill reduction measures on all roads, but if that is not possible, it would be desirable to apply measures based on the priority of the areas in need (e.g., persistent hotspots or intensifying hotspots, etc.). This study thus conducted nationwide expressway hotspot analysis by animal species over 16 years; in particular, it introduced spatiotemporal analysis for the first time in Korea. This study characterized the distribution of eight types of hotspots and eight types of cold spots based on their trends over time and identified high- and low-risk areas for roadkill using spatiotemporal trend analysis. Using this information, it is possible to establish specific and priority measures for these hotspot/cold spot types in order to reduce WVCs. In particular, it is suggested that strategies are needed to prevent collisions with water deer in persistent and consecutive hotspot areas, while accidents involving wild boar need to be addressed in new hotspots, mainly in the capital area and parts of Chungnam province and Gyeongbuk province. Therefore, the results of this study are expected to contribute greatly to the establishment of roadkill reduction measures at the national level for individual animal species.

## Figures and Tables

**Figure 1 ijerph-20-04896-f001:**
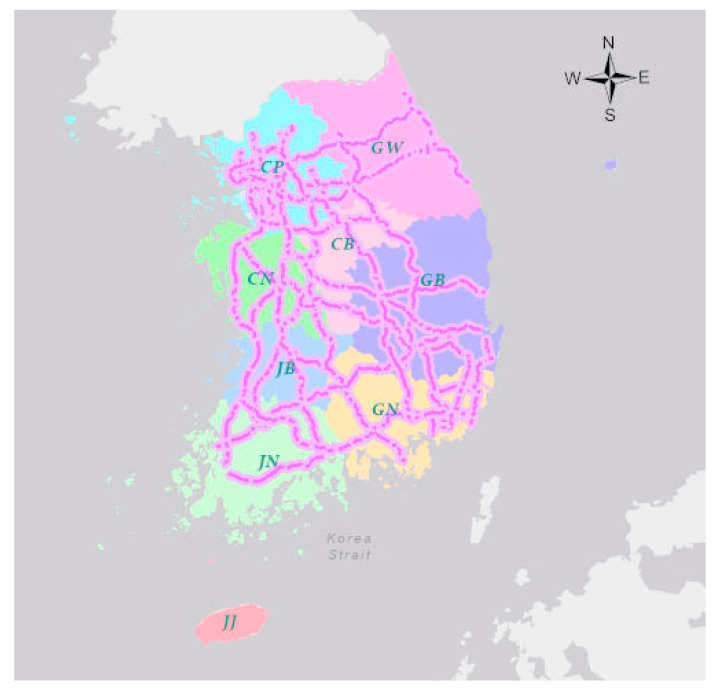
Map of the expressways and the eight administrative provinces in Korea. CP: Capital area, GW: Gangwon province, CN: Chungnam province, CB: Chungbuk province, JB: Jeonbuk province, GB: Gyeongbuk province, JN: Jeonnam province, GN: Gyeongnam province, JJ: Jeju-do.

**Figure 2 ijerph-20-04896-f002:**
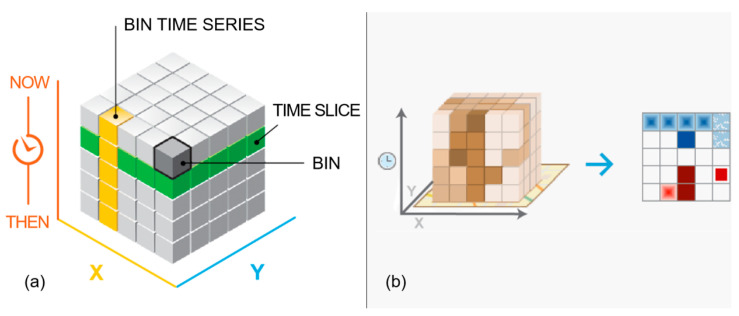
Structure of space–time cube (http://pro.arcgis.com). (**a**) Space–time bins in 3D; (**b**) Generated bins 2D for emerging hotspot analysis.

**Figure 3 ijerph-20-04896-f003:**
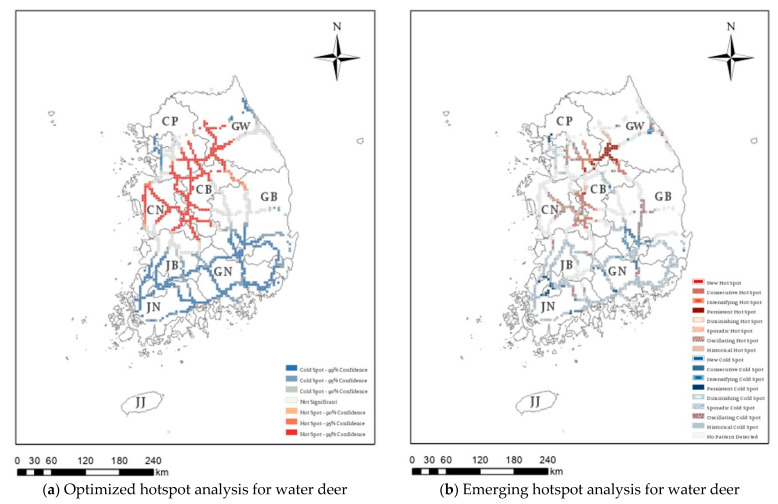
Optimized hotspot analysis (**a**) and emerging hotspot analysis (**b**) for water deer (*Hydropotes inermis*) roadkill on Korean expressways between 2004 and 2019. CP: Capital area, GW: Gangwon province, CN: Chungnam province, CB: Chungbuk province, JB: Jeonbuk province, GB: Gyeongbuk province, JN: Jeonnam province, GN: Gyeongnam province, JJ: Jeju-do.

**Figure 4 ijerph-20-04896-f004:**
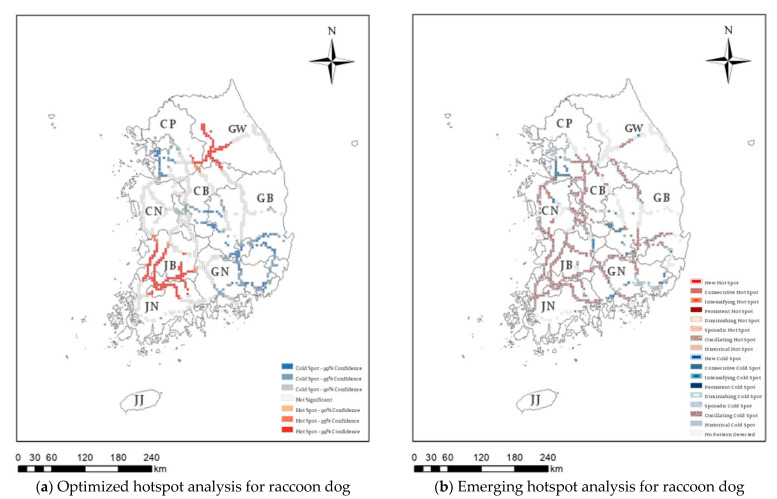
Optimized hotspot analysis (**a**) and emerging hotspot analysis (**b**) for raccoon dog (*Nyctereutes procyonoides*) roadkill on Korean expressways between 2004 and 2019. CP: Capital area, GW: Gangwon province, CN: Chungnam province, CB: Chungbuk province, JB: Jeonbuk province, GB: Gyeongbuk province, JN: Jeonnam province, GN: Gyeongnam province, JJ: Jeju-do.

**Figure 5 ijerph-20-04896-f005:**
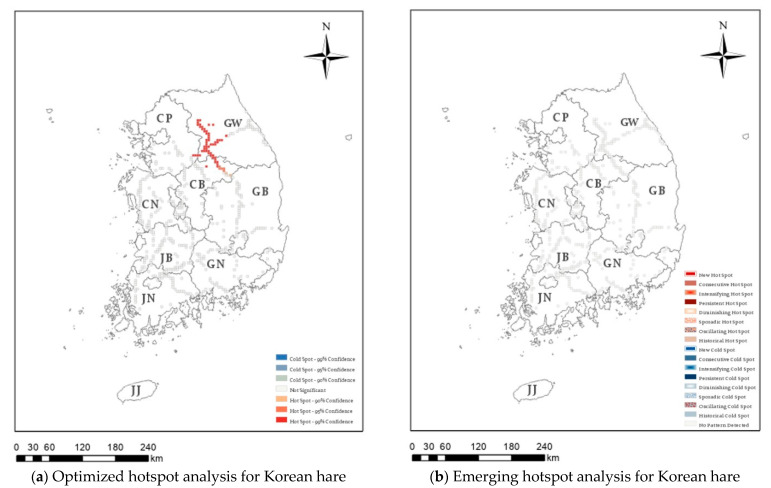
Optimized hotspot analysis (**a**) and emerging hotspot analysis (**b**) for Korean hare (*Lepus coreanus*) roadkill on Korean expressways between 2004 and 2019. CP: Capital area, GW: Gangwon province, CN: Chungnam province, CB: Chungbuk province, JB: Jeonbuk province, GB: Gyeongbuk province, JN: Jeonnam province, GN: Gyeongnam province, JJ: Jeju-do.

**Figure 6 ijerph-20-04896-f006:**
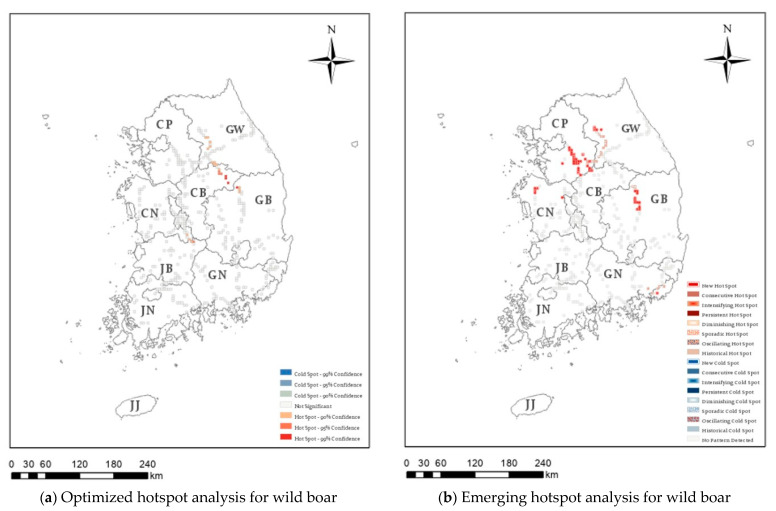
Optimized hotspot analysis (**a**) and emerging hotspot analysis (**b**) for wild boar (*Sus scrofa*) roadkill on Korean expressways between 2004 and 2019. CP: Capital area, GW: Gangwon province, CN: Chungnam province, CB: Chungbuk province, JB: Jeonbuk province, GB: Gyeongbuk province, JN: Jeonnam province, GN: Gyeongnam province, JJ: Jeju-do.

**Figure 7 ijerph-20-04896-f007:**
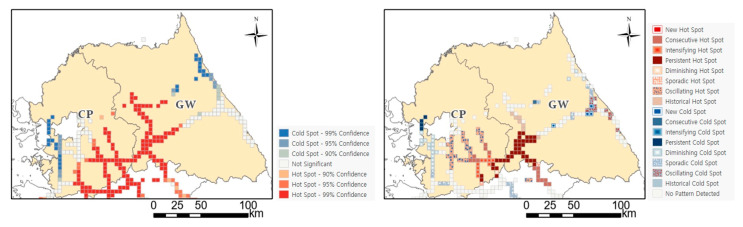
Optimized hotspot and Emerging hotspot of water deer roadkill in Gangwon Province and Capital area. CP: Capital area, GW: Gangwon province.

**Table 1 ijerph-20-04896-t001:** Definition of the types of hotspot and cold spot identified in the spatiotemporal analysis (http://pro.arcgis.com).

	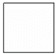	No Pattern Detected	Does Not Fall into Any of the Patterns Described Below.
Hotspots	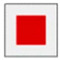	New Hotspot	A location that is a statistically significant hotspot in the final time period and has not been a statistically significant hotspot before.
	Consecutive Hotspot	A location with a single uninterrupted run of statistically significant hotspots in the final time periods. The location has not been a statistically significant hotspot prior to the final hotspot run and fewer than 90% of all time periods are statistically significant hotspots.
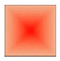	Intensifying Hotspot	A location that has been a statistically significant hotspot for 90% of the time periods, including the final period. In addition, the intensity of the counts for each time period is increasing overall, and this increase is statistically significant.
	Persistent Hotspot	A location that has been a statistically significant hotspot for 90% of the time periods with no discernible increase or decrease in intensity over time.
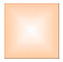	Diminishing Hotspot	A location that has been a statistically significant hotspot for 90% of the time periods, including the final time period. In addition, the intensity in each time period is decreasing overall, and this decrease is statistically significant.
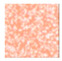	Sporadic Hotspot	A location that is an on-again, off-again hotspot. Less than 90% of the time periods were statistically significant hotspots and none were statistically significant cold spots.
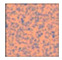	Oscillating Hotspot	A statistically significant hotspot for the final period that has a history of also being a statistically significant cold spot during a prior time period. Less than 90% of the time periods are statistically significant hotspots.
	Historical Hotspot	The most recent time period is not a hot spot, but at least 90% of the periods are statistically significant hotspots.
Cold spot	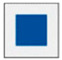	New Cold Spot	A location that is a statistically significant cold spot for the final time period and has not been a statistically significant cold spot before.
	Consecutive Cold Spot	A location with a single uninterrupted run of statistically significant cold spots in the final time periods. The location has not been a statistically significant cold spot prior to the final cold spot run and less than 90% of all time periods are statistically significant cold spots.
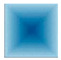	Intensifying Cold Spot	A location that has been a statistically significant cold spot for 90% of the time periods, including the final period. In addition, the intensity of the cold spots is increasing overall and this increase is statistically significant.
	Persistent Cold Spot	A location that has been a statistically significant cold spot for 90% of the time periods with no discernible increase or decrease in the intensity of the count over time.
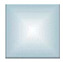	Diminishing Cold Spot	A location that has been a statistically significant cold spot for 90% of the time periods, including the final period. In addition, the intensity of the cold spots in each period is decreasing overall and this decrease is statistically significant.
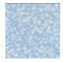	Sporadic Cold Spot	A location that is an on-again, off-again cold spot. Less than 90% of the time periods are statistically significant cold spots and none are statistically significant hotspots.
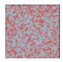	Oscillating Cold Spot	A statistically significant cold spot for the period that has a history of also being a statistically significant hotspot during a prior time period. Less than 90% of the time periods are statistically significant cold spots.
	Historical Cold Spot	The most recent time period is not cold, but at least 90% of the time periods are statistically significant cold spots.

**Table 2 ijerph-20-04896-t002:** Number of cells (4 × 4 km) by type derived by emerging hotspot analysis.

Species	Water Deer		Raccoon Dog		Korean Hare		Wild Boar	
Type	Hot	Cold	Hot	Cold	Hot	Cold	Hot	Cold
New	0	4	0	10	0	0	49	0
Consecutive	49	36	0	44	0	0	0	0
Intensifying	0	24	0	0	0	0	0	0
Persistent	41	20	0	0	0	0	0	0
Diminishing	1	20	0	0	0	0	0	0
Sporadic	9	339	0	65	0	0	25	0
Oscillating	78	57	0	506	0	0	0	0
Historical	7	0	0	0	0	0	0	0

## Data Availability

Not applicable.

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
