# Peer review of "Identification of Emerging Roadkill Hotspots on Korean Expressways Using Space–Time Cubes"

_ijerph, 2023, doi:10.3390/ijerph20064896_

Round 1

Reviewer 1 Report

Dear Authors, the manuscript has a lot of potential due to the huge database regarding road kill in your country and also the time span for collecting it. It is necessary to improve all the manuscript and use software which can really put a high value on these interesting data sets. It is also needed to add new data regarding the densities of the species, and also by the near surrounding habitat/land use near the roads.  

Additional Comments

• What is the main question addressed by the research?

The main Question addressed by the research is “what are the hotspots for roadkill in Korea?”

• Do you consider the topic original or relevant in the field? Does it

address a specific gap in the field?

The topic is very relevant in the field because roadkill are a big threat worldwide for wildlife species and also for the drivers. Every year millions of individuals of different species (including rare/threatened species) are killed. It does not address a gap in the field taking into consideration the method used and especially the results obtained.

• What does it add to the subject area compared with other published

material?

It does not add anything to the subject compared with other published material, except the regional scape (some data from Korea).

• What specific improvements should the authors consider regarding the

methodology? What further controls should be considered?

It is needed to add new data regarding the densities of the target species, the surrounding habitat/land use near the roads, the number of vehicles which are using the roads (especially in the night, when most of the target species are active) etc.

• Are the conclusions consistent with the evidence and arguments presented

and do they address the main question posed?

• Are the references appropriate?

• Please include any additional comments on the tables and figures.

Figures 3-6 – looks more like “long spots”, than hotspots. The Figures are basically a summary of the data regarding road kills, and not a detailed presentation of something that can actually be used by the stakeholders in order to mitigate the conflicts(roadkill) in the area.

Author Response

Attached file

Reviewer 2 Report

The roadkill data in this study should be clearly described in detail.

NB regression models are usually used to identify the hotspot locations. Thus, how to validate the findings from this study.

The paper writing should be improved.

Line 251 to Line 277 should be revised.

More analysis should be conducted.

Some studies on wildlife crash analysis should be reviewed. For example, see: Comparative analysis of the reported animal-vehicle collisions data and carcass removal data for hotspot identification. Journal of advanced transportation, 2019. A copula-based approach for accommodating the underreporting effect in wildlifevehicle crash analysis. Sustainability, 11(2), 418. Investigating the Safety Effectiveness of Wildlife–Vehicle Crash Countermeasures using a Bayesian Approach with a Comparison between Carcass Removal Data and Traditional Crash Data. Transportation Research Record, 03611981221083904.

How to consider the underreporting issue in the hotspot identification.

Round 2

Reviewer 1 Report

Dear Authors,

At the results, from line 169 to line 174, in the new submitted manuscript, you have different data. Why  did you changed the results, taking into consideration that you used the same method and the same data?

Author Response

  1. At the results, from line 169 to line 174, in the new submitted manuscript, you have different data. Why did you changed the results, taking into consideration that you used the same method and the same data?
  • Thank you very much for your thoughtful comments. It was revised according to the opinions of the reviewers, and the results were revised by simply unifying the projection without changing the method. Thanks again for your comments.

Reviewer 2 Report

The comments are well addressed.

Author Response

Thank you for your comments